# Efficacy and Safety of Polaprezinc-Based Therapy versus the Standard Triple Therapy for *Helicobacter pylori* Eradication: A Systematic Review and Meta-Analysis of Randomized Controlled Trials

**DOI:** 10.3390/nu14194126

**Published:** 2022-10-04

**Authors:** Abdelrahman Mahmoud, Mohamed Abuelazm, Ali Ashraf Salah Ahmed, Hassan Abdalshafy, Basel Abdelazeem, James Robert Brašić

**Affiliations:** 1Faculty of Medicine, Minia University, Minia 61519, Egypt; 2Faculty of Medicine, Tanta University, Tanta 31527, Egypt; 3Faculty of Medicine, Cairo University, Cairo 11562, Egypt; 4Department of Internal Medicine, McLaren Health Care, Flint, MI 48532, USA; 5Department of Internal Medicine, Michigan State University, East Lansing, MI 48823, USA; 6Section of High Resolution Brain Positron Emission Tomography Imaging, Division of Nuclear Medicine and Molecular Imaging, The Russell H. Morgan Department of Radiology and Radiological Science, The Johns Hopkins University School of Medicine, Baltimore, MD 21287, USA

**Keywords:** alternative intervention, carnosine, confidence interval, flow chart, gastrointestinal disorder, placebo, protocol, random, treatment, zinc

## Abstract

*Helicobacter pylori* (*H. pylori*) is the most prevalent etiology of gastritis worldwide. *H. pylori* management depends mainly on antibiotics, especially the triple therapy formed of clarithromycin, amoxicillin, and proton pump inhibitors. Lately, many antibiotic-resistant strains have emerged, leading to a decrease in the eradication rates of *H. pylori.* Polaprezinc (PZN), a mucosal protective zinc-L-carnosine complex, may be a non-antibiotic agent to treat *H. pylori* without the risk of resistance. We performed a systematic review and meta-analysis to evaluate the efficacy and safety of a PZN-based regimen for the eradication of *H. pylori.* This study used a systematic review and meta-analysis synthesizing randomized controlled trials (RCTs) from WOS, SCOPUS, EMBASE, PubMed, and Google Scholar until 25 July 2022. We used the odds ratio (OR) for dichotomous outcomes presented with the corresponding 95% confidence interval (CI). We registered our protocol in PROSPERO with ID: CRD42022349231. We included 3 trials with a total of 396 participants who were randomized to either PZN plus triple therapy (*n* = 199) or triple therapy alone (control) (*n* = 197). Pooled OR found a statistical difference favoring the PZN arm in the intention to treat and per protocol *H. pylori* eradication rates (OR: 2.01 with 95% CI [1.27, 3.21], *p =* 0.003) and (OR: 2.65 with 95% CI [1.55, 4.54], *p =* 0.0004), respectively. We found no statistical difference between the two groups regarding the total adverse events (OR: 1.06 with 95% CI [0.55, 2.06], *p =* 0.85). PZN, when added to the triple therapy, yielded a better effect concerning the eradication rates of *H. pylori* with no difference in adverse event rates, and thus can be considered a valuable adjuvant for the management of *H. pylori.* However, the evidence is still scarce, and larger trials are needed to confirm or refute our findings.

## 1. Introduction

*Helicobacter pylori* (*H. pylori*), a virulent Gram-negative organism infecting mainly the human gastric mucosa, afflicted nearly 4.4 billion of the world’s population in 2015 [1]. Chronic infection with *H. pylori* can lead to the emergence of some serious alimentary complications, such as chronic gastritis, irritable bowel syndrome, peptic ulcer, and gastric cancer, the third most prevalent etiology of cancer-associated mortality around the world, ending the lives of over 850,000 humans every year [2,3,4,5]. In particular, *H. pylori* infection is associated with multiple possible etiologies of irritable bowel syndrome, including post-infectious responsiveness, inflammation, and alteration of the gut microorganisms [3,6,7]. Accepted extra-gastric manifestations of *H. pylori* infection are iron deficiency anemia, immune thrombocytopenic purpura, and vitamin B12 deficiency [8]. Moreover, it may increase the risk of acute coronary syndrome [9,10], cerebrovascular disease [11], and neurodegenerative diseases such as Parkinson’s disease and Alzheimer’s disease [12,13]. The eradication of *H. pylori* plays a key role in decreasing the incidence of these complications.

The current frontline recommended regimen includes typical triple therapy (proton pump inhibitor (PPI), clarithromycin, and amoxicillin or metronidazole) or bismuth-based quadruple therapy (PPI or H2 receptor antagonists, metronidazole, tetracycline, and bismuth) and other antibiotic-based options [14,15,16]. With the global development of antibiotic resistance, the diminished efficacy of clarithromycin, metronidazole, and levofloxacin is reaching an alarming level of 15% [17,18,19].

Therefore, we need to widen our scope, find new innovative solutions, and decrease our dependence on antibiotics. Variable gastric mucosal protective agents have been proposed to help in peptic ulcer healing and in the eradication of *H. pylori*, such as rebamipide [20], sofalcone [21], and sucralfate [22], which have the advantage of being unaffected by drug resistance.

Moreover, polaprezinc (PZN), a zinc-L-carnosine complex (Figure 1) [23], has promising properties as an antioxidant promoting ulcer healing and mucosal protective agent to counteract various clinical conditions in animals and human studies [24,25,26,27,28,29,30,31,32]. Furthermore, PZN can function by ameliorating inflammation [33], preventing apoptosis [34], and protecting tight junctions. Additionally, PZN was reported to decrease the indomethacin-induced increase in the gut permeability [35,36], indicating a small bowel protective effect [37,38]. Vascularly, PZN was reported to activate the mesenchymal stem cells and increase the expression of insulin-like growth factor in endothelial tissue protecting the injured gastric and skin lesions [39,40]. Accordingly, PZN is a promising agent that can be implemented within the *H. pylori* treatment protocol; however, strong synthesized evidence is still lacking. Hence, our study’s goal is to assess the effectiveness and safety of PZN as a supportive agent to triple therapy (PPI + clarithromycin + amoxicillin) for the management of patients with *H. pylori* infection.

## 2. Materials and Methods

### 2.1. Protocol Registration

Our review was prospectively registered and published in an international prospective register of health-related systematic reviews (PROSPERO) with ID: CRD42022349231. We performed a systematic review and meta-analysis in accordance with the Preferred Reporting Items for Systematic Reviews and Meta-Analyses (PRISMA) statement [41,42,43] and the Cochrane Handbook of Systematic reviews and meta-analysis [44]. The process is documented in a PRISMA 2020 checklist (Appendix A).

### 2.2. Data Sources and Search Strategy

Web of Science, SCOPUS, EMBASE, PubMed (MEDLINE), Google Scholar, and Cochrane Central Register of Controlled Trials (CENTRAL) were comprehensively searched by two reviewers (A.M. and M.A.) until 25 July 2022. We used no filters. The thorough selection procedure is illustrated in (Table 1).

### 2.3. Eligibility Criteria

We included randomized controlled trials (RCTs) with the following PICO criteria: population (P): patients with *H. pylori* infection; intervention (I): PZN 150 mg plus triple therapy (amoxicillin, clarithromycin, and PPI), control (C) triple therapy only and outcome (O): the primary outcome of this study is to evaluate the eradication rate of *H. pylori* (patients who achieved *H. pylori* clearance) according to intention to treat or per protocol analysis. The secondary outcome is the safety, defined as any reported adverse events. The exclusion criteria involved animal studies, cohort, retrospective, case reports, case reports, non-randomized trials, laboratory studies, and conference abstracts.

### 2.4. Study Selection

After duplicates removal using the Covidence online tool [45], two investigators (A.M. and H.A.) independently checked the eligibility of titles and abstracts of the obtained records. Then, they evaluated the full texts of the relevant studies according to the previously mentioned eligibility criteria. Any discrepancies were solved via discussion to reach a consensus.

### 2.5. Data Extraction

Using a pilot-tested extraction form, two reviewers (A.A.S.A. and H.A.) separately extracted the following data from the included articles: study characteristics (year of publication, country, study design, total participants, used triple therapy, frequency, and dose of PZN and method by which *H. pylori* was diagnosed); baseline information (age, sex, number of patients in each group, and number and location of ulcers); and efficacy outcomes data (intention-to-treat *H. pylori* eradication rate, per-protocol *H. pylori* eradication rate, and adverse events including (nausea, vomiting, heartburn, diarrhea, skin rash, and total adverse events). Disagreements were resolved by another investigator (A.M.).

### 2.6. Risk of Bias and Quality Assessment

The Cochrane Collaboration’s technique was our guide to evaluate the risk of bias in randomized trials; two reviewers (A.A.S.A. and H.A.) separately evaluated the included studies for risk of bias (ROB) [46], based on the following six items: random sequence generation (selection bias), allocation concealment (selection bias), blinding of participants and personnel (performance bias), blinding of outcome assessment (detection bias), incomplete outcome data (attrition bias), selective reporting (reporting bias), and other potential sources of bias. Disagreements were settled through discussion. Two reviewers (M.T. and B.A.) employed the Grading of Recommendations Assessment, Development, and Evaluation (GRADE) guidelines to appraise the quality of the evidence [47,48,49]. Imprecision, indirectness, inconsistency, publication bias, and bias risk were evaluated. Our results about the quality of evidence were justified, written, and included in each outcome. Any discrepancies were handled through discussion.

### 2.7. Statistical Analysis

The statistical analysis was carried out with Revman software version 5.4 [50]. We used odds ratio to pool dichotomous outcomes presented with the corresponding 95% confidence interval (CI). We utilized the I-square and Chi-square tests to assess heterogeneity; while the Chi-square test tells whether there is heterogeneity, the I-square determines the depth of heterogeneity. A grand heterogeneity (for the Chi-square test) is named as an alpha level below 0.1, in accordance with the Cochrane Handbook (chapter nine) [46], while the I-square test is interpreted as: (0–40 percent: not significant; 30–60 percent: moderate heterogeneity; 50–90 percent: substantial heterogeneity). We used the fixed-effects model. We calculated the number needed to treat (NNT) via the next equation, Absolute risk reduction (ARR) = (control event rate) − (experimental event rate) and the NNT equals the inverse of the ARR.

## 3. Results

### 3.1. Search Results and Study Selection

We identified 1496 records after searching the databases, then 635 duplicates were excluded. Title and abstract screening excluded 841 irrelevant records. We moved to full-text screening with 20 articles, and 17 articles were excluded. Finally, only three articles met our inclusion criteria. The PRISMA flow chart of the detailed selection process is demonstrated in (Figure 2).

### 3.2. Characteristics of Included Studies

We included 3 trials with a total of 396 participants who were randomized to either PZN plus triple therapy (*n* = 199) or triple therapy alone (control) (*n* = 197). Further included trials’ characteristics are presented in (Table 2). PZN dose was 150 mg twice daily for seven days in two trials [26,27] and for fourteen days in one trial [28]. Male participants were a total of 122 (61.3%) in the PZN group and 124 (62.94%) in the control group. Further baseline characteristics of the participants are presented in (Table 3).

### 3.3. Risk of Bias and Quality of Evidence

We appraised the quality of the included studies according to the Cochrane risk of bias tool [46], as shown in Figure 3. Regarding the selection bias, Isomoto et al. [26] had low risk in the random sequence generation and unclear risk in the allocation concealment, Kashimura et al. [27] had unclear risk in both domains, and Tan et al. [28] had low risk in both domains. Moreover, the included trials had a high risk of performance and detection biases, except Kashimura et al. [27], with a low risk of performance and detection biases. Additionally, the included trials had a low risk of attrition bias. Furthermore, all included trials had an unclear risk of reporting bias. Finally, the included trials had a low risk of other bias. Author judgments are furtherly clarified in the Appendix (Appendix B). Using the GRADE system, the included primary outcomes yielded very-low-quality evidence. Details and explanations are clarified in Table 4.

### 3.4. Primary Outcomes

#### 3.4.1. *H. pylori* Eradication Rates Based on Intention to Treat Analysis

The pooled analysis favored the PZN group (OR: 2.01 with 95% CI [1.27, 3.21], *p =* 0.003) (very-low-quality evidence) (Figure 4A, Table 4). The pooled studies were homogenous (*p =* 0.27, I-square = 24%). From our calculation of the NNT on average, 7.5 patients would have to receive PZN treatment (instead of control treatment) for one additional patient to have the outcome, ARR = 0.67 − 0.804 = − 0.134. NNT = 1/ARR = 1/− 0.134 = −7.5.

#### 3.4.2. *H. pylori* Eradication Rates Based on per Protocol Analysis

The pooled analysis favored the PZN group (OR: 2.65 with 95% CI [1.55, 4.54], *p* = 0.0004) (very-low-quality evidence) (Figure 4B, Table 4). The pooled studies were homogenous (*p* = 0.21, I-square = 36%). On average, 6.3 patients would have to receive PZN treatment (instead of control treatment) for one additional patient to have the outcome, ARR = 0.70.88 − 0.86.81 = −0.1593. NNT = 1/ARR = 1/−0.1593 = −6.3.

### 3.5. Secondary Outcomes

#### 3.5.1. Total Patients with Adverse Events

We found no difference between the two groups (OR: 1.06 with 95% CI [0.55, 2.06], *p* = 0.85) under the fixed-effects model (very-low-quality evidence). The pooled studies were homogenous (*p* = 0.35, I-square = 5%) (Figure 5A).

#### 3.5.2. Specific Adverse Events

Only two trials, Isomoto et al. [26] and Kashimura et al. [27], reported specific adverse events incidence, and we found no difference between the two groups regarding the incidence of diarrhea (OR: 1.19 with 95% CI [0.54, 2.66], *p* = 0.67), vomiting or nausea (OR: 0.32 with 95% CI [0.01, 8.06], *p* = 0.49), and rash (OR: 0.93 with 95% CI [0.13, 6.66], *p* = 0.95) (Figure 5B).

## 4. Discussion

*H. pylori* infection and colonization of the human gastric mucosa are prevalent in over 50% of the world’s population [51]. Although most cases are asymptomatic, *H. pylori* can lead to significant complications, including peptic ulcer disease, gastric adenocarcinoma, and mucousa-associated lymphoma [52,53] Specifically, the incidence of peptic ulcer disease is about 10 to 20% of *H. pylori* patients with about 1 to 3% cases complicated by gastric cancer [4]. Accordingly, the burden of *H. pylori* is overwhelming, and an effective *H. pylori* eradication strategy is required. Therefore, we evaluated the efficacy and safety of PZN as an adjuvant muco-protective agent in adjuvant with the standard triple therapy to eradicate *H. pylori*.

Regarding the *H. pylori* eradication rate, our pooled analysis favored PZN over triple therapy alone in both ITT analysis (80.4% versus 67.01%) and per-protocol analysis (86.8% versus 70.9%), respectively. Moreover, the incidence of adverse events was similar in both groups.

The specific mechanism of the PZN role in enhancing the eradication of *H. pylori* is still to be investigated, with several proposed theories: first, zinc can inhibit the urease activity leading to *H. pylori*’s growth retardation by replacing the nickel ions at the active site of urease hindering the two metal ions from the complex formation [54]. Second, zinc can decrease the expression of interleukin 1 beta (IL-1β) by the gastric mucosa, further inhibiting *H. pylori* growth [55]. Third, PZN has shown to scavenge the monochloramine in *H. pylori*-infected Mongolian gerbils [25]. Finally, zinc has been shown to form a complex with famotidine inhibiting the urease enzyme and, subsequently, *H. pylori* growth, which was evident in both the antibiotic-resistant and sensitive strains [56]

Recently, in comparative transcriptome analysis, Fan et al. proposed multiple potential anti-*H. pylori* effects of zinc [57]. First, zinc can alter the composition, structure, and function of the *H. pylori* type IV secretion system by the downregulation of cagI gene; hence, zinc can partially block the pathogenicity of *H. pylori.* Second, zinc can alter the synthesis process of lipopolysaccharide (LPS), a significant virulent factor of *H. pylori,* by altering the biosynthesis of lipid A (a significant hydrophobic part of LPS). *H. pylori*’s surface LPS is a significant part of its cell wall contributing to the adhesion and infection of the gastric mucosa [57,58]. Therefore, disrupting LPS synthesis can subsequently affect the infectivity and adaptability of *H. pylori* [57]. Third, zinc upregulated the *H. pylori* translation and transcription genes, subsequently leading to increased protein biosynthesis, which can be an adaptation mechanism of *H. pylori;* however, Fan et al. argue that the synthesis of large amounts of in vivo proteins without the help of enough chaperones can lead to accumulation of mis- and unfolded proteins, subsequently disturbing the proteostasis and hindering *H. pylori* growth and even cell death [57]. Finally, zinc disrupted the flagellar protein assembly, disrupting *H. pylori* cell motility [57].

Regarding the status of high antimicrobial agents’ resistance, implementing PZN into *H. pylori* can be beneficial. To clarify, the H. pylori resistance to clarithromycin and metronidazole is currently reported to be ≥ 15% [18,19], leading to a significant drop in the *H. pylori* eradication rates of triple therapy between 50% and 70% [18,19], which is significantly lower than the recommended ITT Maastricht *H. pylori* eradication rate of >80% [15]. Accordingly, PZN regimen can be effectively used for *H. pylori* with an ITT *H. pylori* eradication rate of 80.4%. Moreover, in a recent RCT, PZN was adjunctly used with the bismuth quadruple therapy achieving an *H. pylori* eradication rate of 93.5%, which was statistically significant in comparison with the triple therapy [24].

Regarding safety, PZN was safe and well tolerable in comparison with the triple therapy. The typical PZN dose is 150 mg, containing 34 mg zinc and 116 mg L-carnosine [59]. All the included trials used the typical dose with no crucial adverse events, and the reported adverse events were minor and faded spontaneously or managed feasibly [26,27,28]. However, Tan et al. observed more adverse events associated with the high-dose PZN (300 mg); they attributed this effect to either the toxic effect of the high dosage or patients’ self-hypersensitivity [28]. Accordingly, the standard dose of PZN (150 mg) can be used safely with triple therapy.

### 4.1. Strengths

To the best of our awareness, this is the first systematic review and meta-analysis synthesizing evidence on the efficacy and safety of PZN for *H. pylori* eradication; hence, this study constitutes gold standard evidence in this regard. Moreover, our review was executed and fulfilled via the guidance of the PRISMA recommendations [42,43].

### 4.2. Limitations

Our review has a few limitations. First, we only included three RCTs with a small sample size and limited population distribution confined to the Far East [26,27,28]. Second, the proton pump inhibitor component of the triple therapy varied across the included trials; hence, this can affect our findings. Third, multiple confounding variables can significantly affect our findings, including smoking habits, genetic predisposition of cytochrome p450 2C19, the physical status of the participants, and *H. pylori* strain resistance. Fourth, all the included trials had a relatively short follow-up duration ranging from one to two months only [26,27,28]. Finally, the GRADE assessment yielded very-low-quality evidence; hence, the extrapolation and the generalization of our findings is limited.

### 4.3. Implications for Future Research

Future trials are required to address: first, the comparative efficacy of PZN adjunctly with the bismuth quadruple therapy versus the bismuth quadruple therapy alone is still to be investigated. To clarify, bismuth quadrable therapy is currently recommended as the first-line regimen in areas with a significant prevalence of ciprofloxacin and metronidazole resistance. As such, investigating the efficacy of PZN in the settings with significant resistance is still required [60]. Second, future trials should determine the baseline clarithromycin resistance to enable health authorities to predict the *H. pylori* eradication rate of PZN-based regimen in areas with known rates of clarithromycin resistance using the H. pylori-nomogram [28,61]. Finally, future trials should expand the follow-up duration up to 6 or 12 months to properly investigate the improvement in the gastrointestinal symptoms [28].

## 5. Conclusions

The addition of PZN to the triple therapy yielded greater eradication rates of *H. pylori* with no difference in adverse event rates and thus constitutes a valuable adjuvant for the management of *H. pylori.* However, the evidence is still scarce, and larger trials are needed to confirm or refute our findings. As such more high-quality, multicenter randomized controlled trials are warranted to ascertain its efficacy and yield generalizable findings.

## Figures and Tables

**Figure 1 nutrients-14-04126-f001:**
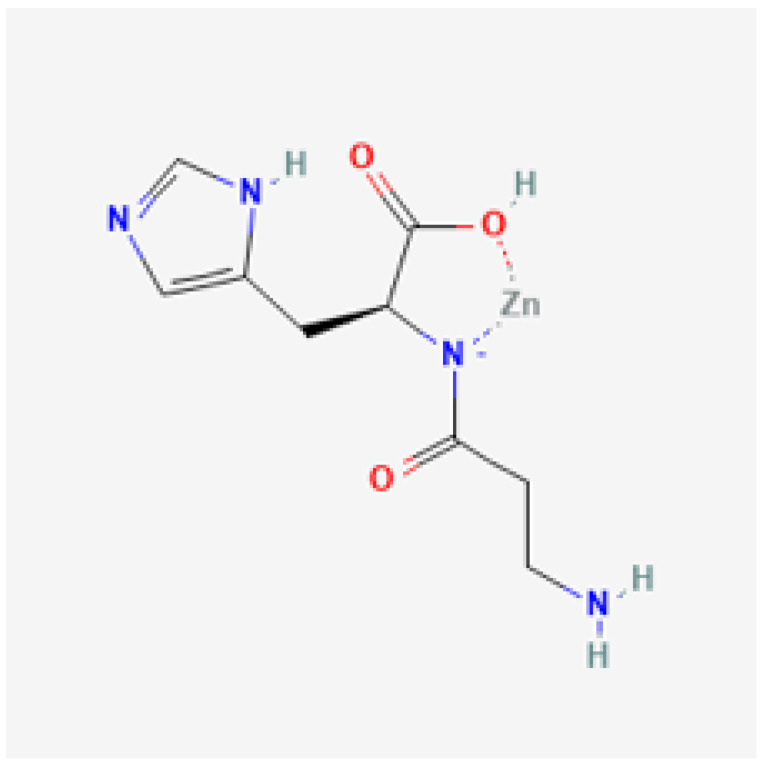
Chemical structure of polaprezinc. Courtesy of the U.S. National Library of Medicine [23].

**Figure 2 nutrients-14-04126-f002:**
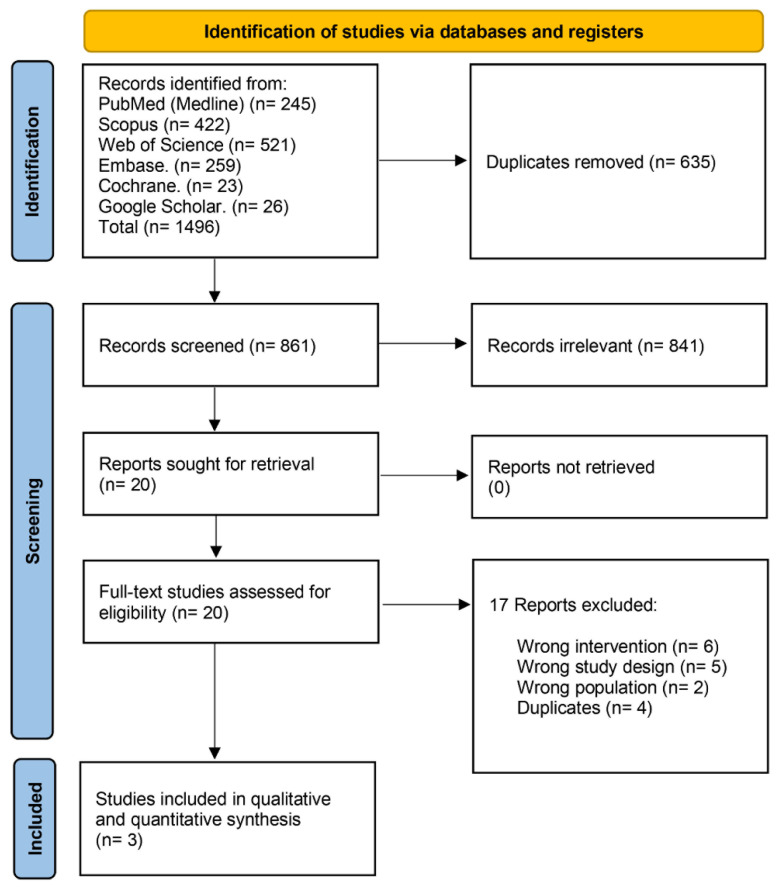
PRISMA flow chart of the screening process.

**Figure 3 nutrients-14-04126-f003:**
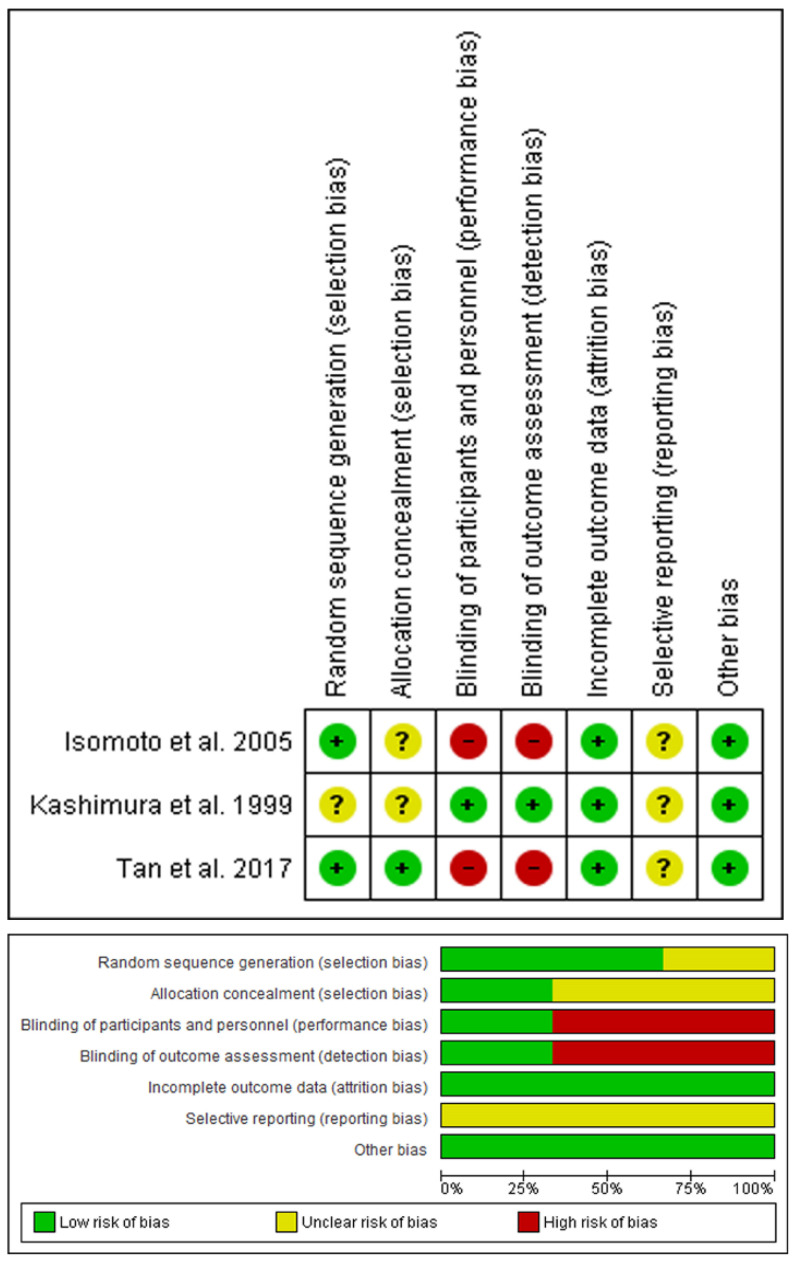
Quality assessment of risk of bias in the studies in the meta-analysis. The upper panel presents a schematic representation of risks (low = red, unclear = yellow, and high = red) for specific types of biases of each of the studies in the review. The lower panel presents risks (low = red, unclear = yellow, and high = red) for the subtypes of biases of the combination of studies included in this review [50].

**Figure 4 nutrients-14-04126-f004:**
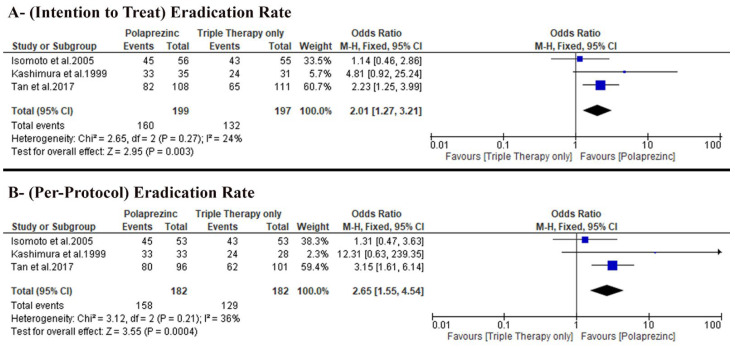
Forest plot of the primary outcome (**A**) *H. pylori* eradication rates based on intention to treat analysis, (**B**) *H. pylori* eradication rate based on per protocol analysis [50]. OR: odds ratio, CI: confidence interval.

**Figure 5 nutrients-14-04126-f005:**
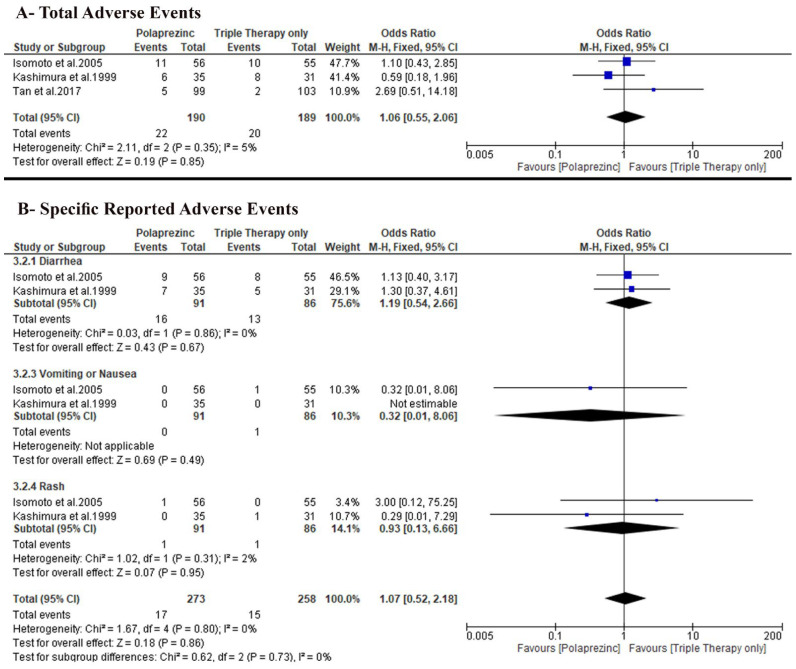
Forest plot of the secondary outcomes (**A**) total adverse events, (**B**) specific reported adverse events [50]. OR: odds ratio, CI: confidence interval.

**Table 1 nutrients-14-04126-t001:** Search terms and results in different databases.

Database	Search Terms	Search Field	Search Results
PubMed	(polaprezinc OR zinc OR zn OR carnosine OR “zinc carnosine”) AND ((*Helicobacter pylori*) OR (*H. pylori*))	All Field	245
Cochrane	(polaprezinc OR zinc OR zn OR carnosine OR “zinc carnosine”) AND ((*Helicobacter pylori*) OR (*H. pylori*))	All Field	23
WOS	(polaprezinc OR zinc OR zn OR carnosine OR “zinc carnosine”) AND ((*Helicobacter pylori*) OR (*H. pylori*))	All Field	521
SCOPUS	(polaprezinc OR zinc OR zn OR carnosine OR “zinc carnosine”) AND ((*Helicobacter pylori*) OR (*H. pylori*))	Title, Abstract,	422
EMBASE	#3. 1 AND #2 #2.‘*Helicobacter pylori’*: ti,ab,kw OR ‘*H. pylori’*: ti,ab,kw #1.zinc: ti,ab,kw OR carnosine: ti,ab,kw OR polaprezinc: ti,ab,kw	All Field	259
Google Scholar	Allintitle: polaprezinc pylori	Allintitle	26

**Table 2 nutrients-14-04126-t002:** Characteristics of the included studies.

Study ID	Study Design	Country	Total Participants	Dose and Frequency of Administration		Method of *H. pylori* Diagnosis
				TT	PZN	
Isomoto et al. [26] 2005	RCT	Single center in China	111	Rabeprazole (10 mg twice daily), clarithromycin (200 mg twice daily) and amoxicillin (750 mg twice daily).	PZN 150 mg twice daily for 7 days	Serology (anti-*H. pylori* immunoglobulin G antibody and histology (Giemsa staining) using two biopsy specimens obtained during endoscopy from each antrum)
Kashimura et al. [27] 1999	RCT	Single center in Japan	66	Lansoprazole 30 mg twice, amoxicillin 500 twice, clarithromycin 400 mg twice for 7 days	PZN 150 mg twice daily for 7 days	Rapid urease test, histology, and culture
Tan et al. [28] 2017	RCT	Single center in China	219	Omeprazole 20 mg, amoxicillin 1 g, and clarithromycin 500 mg, each twice daily	PZN 150 mg twice daily for 14 days	13C or 14C urea breath test and esophagogastroduodenoscopy (EGD)

PZN: polaprezinc, TT: Triple therapy, RCT: randomized controlled trial.

**Table 3 nutrients-14-04126-t003:** Baseline characteristics of the participants.

Study ID	Number of Patients		Age (Year) Mean (Range) Mean ± SD		Gender (Male) N. (%)		Gastric Ulcer N. (%)		Duodenal Ulcer N. (%)		Gastroduodenal Ulcers N. (%)	
	PZN	TT	PZN	TT	PZN	TT	PZN	TT	PZN	TT	PZN	TT
Isomoto et al. [26] 2005	56	55	45.6 (21–71)	45.3 (21–73)	42 (75%)	41 (74.5%)	36 (64.3%)	34 (61.8%)	19 (33.9%)	19 (34.5)	1 (1.8%)	2 (3.6%)
Kashimura et al. [27] 1999	35	31	53.7 (25–70)	55.3 (22–72)	22 (62.8%)	25 (80.6%)	4 (11.4%)	4 (12.9%)	7 (20%)	9 (31%)	2 (2.71%)	2 (6.45%)
Tan et al. [28] 2017	108	111	40.5 ± 13.6	41.0 ± 11.8	58 (53.7)	58 (52.3)	N/A	N/A	N/A	N/A	N/A	N/A

N: number, SD: standard deviation, N/A: not available, PZN: polaprezinc, TT: triple therapy.

**Table 4 nutrients-14-04126-t004:** GRADE evidence profile.

Certainty assessment							№ of Patients		Effect		Certainty	Importance
№ of Studies	Study Design	Risk of Bias	Inconsistency	Indirectness	Imprecision	Other Considerations	Primary Outcome	Placebo	Relative (95% CI)	Absolute (95% CI)		
Intention to treat *H. pylori* eradication rate												
**3**	RCTs	Very serious ^a^	Not serious	Not serious	Serious ^b^	None	160/199 (80.4%)	132/197 (67.0%)	OR 2.01 (1.27 to 3.21)	133 more per 1000 (from 51 more to 197 more)	⨁◯◯◯ Very low	CRITICAL
Per-protocol *H. pylori* eradication rate												
**3**	RCTs	Very serious ^a^	Not serious	Not serious	Serious^b^	None	158/182 (86.8%)	129/182 (70.9%)	OR 2.65 (1.55 to 4.54)	157 more per 1000 (from 82 more to 208 more)	⨁◯◯◯ Very low	CRITICAL

CI: confidence interval; MD: mean difference. ^a^ The included trials showed high risk of bias. ^b^ The total number of events is less than 30.

## Data Availability

All data are included in the manuscript.

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
