# Peer review of "Efficacy and Safety of Polaprezinc-Based Therapy versus the Standard Triple Therapy for Helicobacter pylori Eradication: A Systematic Review and Meta-Analysis of Randomized Controlled Trials"

_nutrients, 2022, doi:10.3390/nu14194126_

Round 1

Reviewer 1 Report

-          Explain better the link between H. pylori and irritable bowel syndrome

-          “Furthermore, many studies link H. pylori to extra-gastroduodenal diseases such as vitamin B12 deficiency, insulin resistance, diabetes mellitus, psoriasis, and non-alcoholic fatty liver disease”

Accepted extra-gastric manifestations of H. pylori infection are iron deficiency anemia, immune thrombocytopenic purpura, and, vitamin B12 deficiency

-          “which is significantly lower than the 241 recommended ITT Maastricht eradication rate of >80% [14].”

Update to new Maastricht VI

-          “Moreover, in a recent RCT, PZN was adjunctly used with the bismuth quadruple therapy achieving an eradication rate of 93.5%”

Cite this RCT

Author Response

Responses to reviewers are indicated in italics.

Reviewer 1: Comments and Suggestions for Authors.

1) -          Explain better the link between H. pylori and irritable bowel syndrome

Thank you for pointing this out. We added to the first paragraph of the introduction we added a sentence as follows:

In particular, H. pylori infection is associated with multiple possible etiologies of irritable bowel syndrome, including post-infectious responsiveness, inflammation, and alteration of the gut microorganisms [3-5].

We added references as follows:

Saha, L. Irritable bowel syndrome: pathogenesis, diagnosis, treatment, and evidence-based medicine. World J. Gastroenterol. 2014, 20(22), 6759-6773. https:// doi.org/10.3748/wjg.v20.i22.6759.PMID: 24944467; PMCID: PMC4051916

Holtmann, G.J.; Ford, A.C.; Talley, N.J. Pathophysiology of irritable bowel syndrome. Lancet Gastroenterol. Hepatol. 2016, 1(2), 133-146. https://doi.org/10.1016/S2468-1253(16)30023-1

2)- change “Furthermore, many studies link H. pylori to extra-gastroduodenal diseases such as vitamin B12 deficiency, insulin resistance, diabetes mellitus, psoriasis, and non-alcoholic fatty liver disease” to “Accepted extra-gastric manifestations of H. pylori infection are iron deficiency anemia, immune thrombocytopenic purpura, and, vitamin B12 deficiency”

Thank you for pointing this out. We changed it accordingly.

3) -“which is significantly lower than the 241 recommended ITT Maastricht eradication rate of >80% [14].” 

Update to new Maastricht VI

Thank you for pointing this out. We updated accordingly as follows:

Malfertheiner, P.; Megraud, F.; Rokkas, T.; Gisbert, J.P.; Liou, J.M.; Schulz, C.; Gasbarrini, A.; Hunt, R.H.; Leja, M.; O’Morain, C.; et al. Management of Helicobacter pylori infection-the Maastricht VI/Florence consensus report. Gut 2022, 71, 1724-1762. https://doi.org/10.1136/gutjnl-2022-327745

4) - “Moreover, in a recent RCT, PZN was adjunctly used with the bismuth quadruple therapy achieving an eradication rate of 93.5%”

Cite this RCT

Thank you for pointing this out. We edited the manuscript accordingly.

Reviewer 2 Report

The manuscript "Efficacy and Safety of Polaprezinc Based Therapy versus the Standard Triple therapy for Helicobacter Pylori Eradication: A Systematic Review and Meta-Analysis of Randomized Controlled Trials" is a well-written comprehensive and systematic review that studies the influence of adding polaprezinc to the triple therapy against Helicobacter pylori. Its main drawback is the very low number of studies covered (only 3 clinical trials) but the treatment is exhaustive and makes the work is worthy to be published in Nutrients, after checking the following minor issues:

1) At page 5, a big blank space occupies almost all the page, due to the inclusion of tables with the opposite orientation in the 6th page. In addition, the introduction, although well written, does not show visually things that could be of interest for ease the reading, as for example adding a scheme with the structures of polaprezin and with the more relevant drugs used in the three studies analyzed. Then, adding this scheme, besides improving the manuscript, it would enable a partial filling of that blank space.

2) Abstract, lines 20-21. Please put "H. pylori" in italics.

3) Just for curiosity, is it not too high the 4.4 billion people affliction data of H. pylori? It supposes more than half of world's population.

4) At Table 4 and onwards, when it is mentioned eradication rate, for clarity, could be better H. pylori eradication rate.

Author Response

The response to reviewer 2 are indicated in italics.

Reviewer 2: Comments and Suggestions for Authors.

The manuscript "Efficacy and Safety of Polaprezinc Based Therapy versus the Standard Triple therapy for Helicobacter Pylori Eradication: A Systematic Review and Meta-Analysis of Randomized Controlled Trials" is a well-written comprehensive and systematic review that studies the influence of adding polaprezinc to the triple therapy against Helicobacter pylori. Its main drawback is the very low number of studies covered (only 3 clinical trials) but the treatment is exhaustive and makes the work is worthy to be published in Nutrients, after checking the following minor issues:

We sincerely thank reviewer 2 for their comments, we really appreciate their time and consideration to improve the manuscript.

1) At page 5, a big blank space occupies almost all the page, due to the inclusion of tables with the opposite orientation in the 6th page. In addition, the introduction, although well written, does not show visually things that could be of interest for ease the reading, as for example adding a scheme with the structures of polaprezin and with the more relevant drugs used in the three studies analyzed. Then, adding this scheme, besides improving the manuscript, it would enable a partial filling of that blank space.

Thank you for pointing this out. We edited the introduction accordingly. We added the chemical structure of polaprezin as Figure 1. In order to express the figures at a readable size, we have placed them on separate pages. There are accompanying white spaces due to our limited skills to format the material. We anticipate that the copy editor with express the document without white spaces.

2) Abstract, lines 20-21. Please put "H. pylori" in italics.

Thank you for pointing this out. We edited it accordingly.

3) Just for curiosity, is it not too high the 4.4 billion people affliction data of H. pylori? It supposes more than half of world's population.

Thank you for pointing this out. The most updated estimated prevalence of H. pylori is 4.4 billion people according to a systematic review synthesizing 184 prevalence reports from 62 countries worldwide.

Hooi, J.K.Y.; Lai, W.Y.; Ng, W.K.; Suen, M.M.Y.; Underwood, F.E.; Tanyingoh. D.; Malfertheiner. P.; Graham, D.Y.; Wong. V.W.S.; Wu. J.Y. et al. Global prevalence of Helicobacter pylori infection: systematic review and meta-analysis. Gastroenterology 2017, 153(2), 420-429. https://doi.org/10.1053/j.gastro.2017.04.022

4) At Table 4 and onwards, when it is mentioned eradication rate, for clarity, could be better H. pylori eradication rate.

Thank you for pointing this out. We edited the manuscript accordingly.